# Neural Program Planner
# for Structured Predictions

**Jacob Biloki**
SayMosaic Inc.
jacob.biloki@mosaix.ai

**Chen Liang**
Google Brain
crazydonkey@google.com

**Ni Lao**
SayMosaic Inc.
ni.lao@mosaix.ai

## Abstract

We consider the problem of improving reinforcement learning (RL) applied to weakly supervised structured prediction (SP) – for example, given a database table and a question, perform a sequence of computation actions on the table, generating a response and a binary success-failure reward signal. This line of research has been successful by leveraging RL to directly optimize the desired metrics of a task, such as the accuracy in question answering or BLEU score in machine translation. However, different from the common RL settings, the environment dynamics are deterministic and known in SP, which haven't been fully utilized by the model-free RL methods that are usually applied. We propose to apply model-based RL methods, which rely on planning as a primary model component. We demonstrate the effectiveness of planning-based SP by applying our Neural Program Planner (NPP) model to an existing SP model, Memory Augmented Policy Optimization (MAPO) Liang et al. (2018). The proposed NPP is optimized to score a given a set of candidate programs from a pretrained search policy, deciding which program is the most promising by utilizing the space searched by the underlying model. NPP is applied after beam creation allowing the space to be observed after execution providing more rich program representation only available to NPP. We evaluate NPP on *weakly supervised program synthesis natural language* (semantic parsing) by evaluating it in both a stacked learning and non stacked learning environment. On the WIKITABLEQUESTIONS benchmark, NPP achieves a new state-of-the-art of $47.2\%$ accuracy.

## 1 Introduction

Numerous results from natural language processing tasks have shown that Structured Prediction (SP) can be cast into a reinforcement learning (RL) framework, and known RL techniques can give formal performance bounds on SP tasks (Daume et al., 2009; Ross et al., 2011; Bengio et al., 2015). RL also directly optimizes task metrics, such as, the accuracy in question answering or BLEU score in machine translation, and avoids the exposure bias problem when compaired to maximum likelihood training that is commonly used in SP (Ross et al., 2011; Ranzato et al., 2016).

However, previous works on applying RL to SP problems often use model-free RL algorithms (e.g., REINFORCE or actor-critic) and fail to leverage the characteristics of SP, which are different than typical RL tasks, e.g., playing video games (Mnih et al., 2015) or the game of Go (Silver et al., 2016b; 2017). In most SP problems conditioned on the input $x$, the environment dynamics, except for the reward signal, is known, deterministic, reversible, and therefore can be searched. This means that there is a perfect model[1] of the environment, which can be used to apply model-based RL methods that utilize planning[2] as a primary model component.

Take semantic parsing (Berant et al., 2013; Pasupat & Liang, 2015) as an example, semantic parsers trained by RL such as Neural Semantic Machine (NSM) (Liang et al., 2017; 2018) typically rely on beam search for inference – the program with the highest probability in beam is used for execution and generating answer. However, the policy, which is used for beam search, may not be

---

[1] A *model* of the environment usually means anything that an agent can use to predict how the environment will respond to its actions (Sutton & Barto, 1998).

[2] *planning* usually refers to any computational process that takes a model as input and produces or improves a policy for interacting with the modeled environment (Sutton & Barto, 1998).

able to assign the highest probability to the correct program. This limitation is due to the policy predicting locally normalized probabilities for each possible action based on the partially generated program, and the probability of a program is a product of these local probabilities.

For example, when applied to the WEBQUESTIONSSP task, NSM made mistakes with two common patterns: (1) the program would ignore important information in the context; (2) the generated program does not execute to a reasonable output, but still receives high probability (spurious programs). Resolving this issue requires using the information of the full program and its execution output to further evaluate its quality based on the context, which can be seen as planning. This can be observed in Figure 4 where the model is asked a question *"Which programming is played the most?"*. The full context of the input table (shown in Table 1) contains programming for a television station. The top program generated by a search policy produces the wrong answer, filtering by a column not relevant to the question. If provided the correct contextual features, and if allowed to evaluate the full program forward and backward through time, we observe that a planning model would be able to better evaluate which program would produce the correct answer.

To handle errors related to context, we propose to train a value function to compute the utility of each token in a program. This utility is evaluated by considering the program and token probability as well as the attention mask generated by the sequence-to-sequence (seq2seq) model for the underlying policy. We also introduce beam and question context with a binary feature representing overlap from question/program and program/program, such as how many programs share a token at a given timestep.

In the experiments, we found that applying a planner that uses a learned value function to re-rank the candidates in the beam can significantly and consistently improve the accuracy. On the WIKITABLEQUESTIONS benchmark, we improve the state-of-the-art by $0.9\%$, achieving an accuracy of $47.2\%$.

## 2 BACKGROUND

### 2.1 WIKITABLEQUESTIONS

WIKITABLEQUESTIONS (Pasupat & Liang, 2015) contains tables extracted from Wikipedia and question-answer pairs about the tables. See Table 1 as an example. There are 2,108 tables and 18,496 question-answer pairs split into train/dev/test set. Each table can be converted into a directed graph that can be queried, where rows and cells are converted to graph nodes while column names become labeled directed edges. For the questions, we use string match to identify phrases that appear in the table. We also identify numbers and dates using the CoreNLP annotation released with the dataset. The task is challenging in several aspects. First, the tables are taken from Wikipedia and cover a wide range of topics. Second, at test time, new tables that contain unseen column names appear. Third, the table contents are not normalized as in knowledge-bases like Freebase, so there are noises and ambiguities in the table annotation. Last, the semantics are more complex comparing to previous datasets like WEBQUESTIONSSP (Yih et al., 2016). It requires multiple-step reasoning using a large set of functions, including comparisons, superlatives, aggregations, and arithmetic operations (Pasupat & Liang, 2015). See (Liang et al., 2018) for more details about the functions.

### 2.2 NEURAL SEMANTIC MACHINE

We adopt the NSM framework open sourced in the Memory Augmented Policy Optimization paper (MAPO) (Liang et al., 2017; 2018), which combines (1) a neural "programmer", which is a seq2seq model augmented by a key-variable memory that can translate a natural language utterance to a program as a sequence of tokens, and (2) a symbolic "computer", which is an Lisp interpreter that implements a domain specific language with built-in functions and provides code assistance by eliminating syntactically or semantically invalid choices. For the Lisp interpreter, it added functions according to (Zhang et al., 2017; Neelakantan et al., 2016) for WIKITABLEQUESTIONS, refer to (Liang et al., 2018) for further detail of the open-sourced implementation. Same as (Liang et al., 2018) we consider the problem of *contextual program synthesis* with weak supervision – i.e., no correct action sequence **a** is given as part of the training data, and training needs to solve the hard problem of exploring a large program space. However, we will focus on improving decision making

Table 1: Wikipedia table for question **nt-3516**: *Which programming is played the most?*

| Call-sign | Location | RF | PSIP | Programming |
|---|---|---|---|---|
| WIVM-LD | Canton | 39 | 39.1 | RTV |
| WIVM-LD | Canton | 39 | 39.2 | TV |
| WIVM-LD | Canton | 39 | 39.3 | PBJ |
| WIVM-LD | Canton | 39 | 39.4 | Faith Ministries Radio ... |
| WIVN-LD | Newcomerstown | 29 | 29.1 | RTV (WIVM-LD Simulcast) |
| WIVN-LD | Newcomerstown | 29 | 29.2 | PBJ |
| WIVN-LD | Newcomerstown | 29 | 29.3 | Tuff TV |
| WIVN-LD | Newcomerstown | 29 | 29.4 | Faith Ministries Radio ... |
| WIVX-LD | Loudonville | 51 | 51.1 | RTV (WIVM-LD Simulcast) |
| WIVD-LD | Newcomerstown | 22 | 22.1 | RTV (WIVM-LD Simulcast) |
| W27DG-D | Millersburg | 27 | 27.1 | RTV (WIVM-LD Simulcast) |

(planning) giving a pretrained search policy, while previous work mainly focus on learning the search policies.

## 3   PROBLEM FORMULATION

The problem of *weakly supervised* contextual program synthesis can be formulated as: generating a program $\mathbf{a}$ by using a parametric mapping function, $\hat{\mathbf{a}} = f_\theta(\mathbf{x})$, where $\theta$ denotes the model parameters. The quality of a generated program $\hat{\mathbf{a}}$ is measured in terms of a scoring or *reward* function $R(\hat{\mathbf{a}}; \mathbf{x}, \mathbf{y})$ with $\mathbf{y}$ being the expected correct answer. The reward function evaluates a program by executing it on a real environment and comparing the emitted output against the correct answer. For example, the reward may be binary that is 1 when the output equals the answer and 0 otherwise. We assume that the context $\mathbf{x}$ includes both a natural language input and an environment, for example an interpreter or a database, on which the program will be executed. Given a dataset of context-answer pairs, $\{(\mathbf{x}_i, \mathbf{y}_i)\}_{i=1}^N$, the goal is to find an optimal parameter $\theta^*$ that parameterizes a mapping of $\mathbf{x} \rightarrow \mathbf{a}$ with maximum empirical return on a *heldout test set*.

In this study we will further decompose the policy $f_\theta$ into the stacking of a *search model* $s_\phi(\mathbf{x})$, which produces a set of candidate programs $\mathcal{B}$ given an environment $\mathbf{x}$, and a *value model* $v_\omega(\mathbf{a}; \mathbf{x}, \mathcal{B})$, which assigns a score $s$ to program $\mathbf{a}$ given the environment and all the candidate programs. Therefore, $\theta = [\phi; \omega]$ and

$$f_\theta(\mathbf{x}) \approx \operatorname*{argmax}_{\mathbf{a} \in s_\phi(\mathbf{x})} v_\omega(\mathbf{a}; \mathbf{x}, s_\phi(\mathbf{x})). \tag{1}$$

The search model $s_\phi$ can be learnt by optimizing a conditional distribution $\pi_\phi(\mathbf{a} \mid \mathbf{x})$ that assigns a probability to each program given the context. That is, $\pi_\phi$ is a distribution over the *countable* set of all possible programs, denoted $\mathcal{A}$. Thus $\forall \mathbf{a} \in \mathcal{A}: \pi_\phi(\mathbf{a} \mid \mathbf{x}) \geqslant 0$ and $\sum_{\mathbf{a} \in \mathcal{A}} \pi_\phi(\mathbf{a} \mid \mathbf{x}) = 1$. Then, to synthesize candidate programs for a novel context, one may find the most likely programs under the distribution $\pi_\phi$ via exact or approximate inference such as beam search. $\mathcal{B} = s_\phi(\mathbf{x}) \approx \operatorname{argmax}_{\mathbf{a} \in \mathcal{A}}^B \pi_\phi(\mathbf{a} \mid \mathbf{x})$. $\pi_\phi$ is typically an *autoregressive* model such as a recurrent neural network: (*e.g.* Hochreiter & Schmidhuber (1997a)) $\pi_\phi(\mathbf{a} \mid \mathbf{x}) \equiv \prod_{i=t}^{|\mathbf{a}|} \pi_\phi(a_t \mid \mathbf{a}_{<t}, \mathbf{x})$, where $\mathbf{a}_{<t} \equiv (a_1, \ldots, a_{t-1})$ denotes a prefix of the program $\mathbf{a}$. In the absence of ground truth programs, policy gradient techniques (such as REINFORCE Williams (1992)) present a way to optimize the parameters of a stochastic policy $\pi_\phi$ via optimization of *expected return*. Given a training dataset of context-answer pairs, $\{(\mathbf{x}^l, \mathbf{y}^l)\}_{l=1}^N$, the training objective is $\mathcal{O}_{\text{ER}}(\theta) = \mathbb{E}_{\mathbf{a} \sim \pi_\phi(\mathbf{a} \mid \mathbf{x})} R(\mathbf{a}; \mathbf{x}, \mathbf{y})$.

Decision-time planning typically relies on value network Silver et al. (2016a) trained to predict the true reward. In the next section, however, we introduce a max-margin training objective for the value model $v_\omega$, which optimizes to rank rewarded programs higher than non-rewarded programs.

## 4   NEURAL PROGRAM PLANNER (NPP)

We now introduce NPP by first describing the architecture of $v_\omega$ – a seq2seq model which goes over candidate program answer pairs and the final score of a candidate program is simply the sum of its token scores (Section 4.1). Secondly we describe the program token representation, which considers

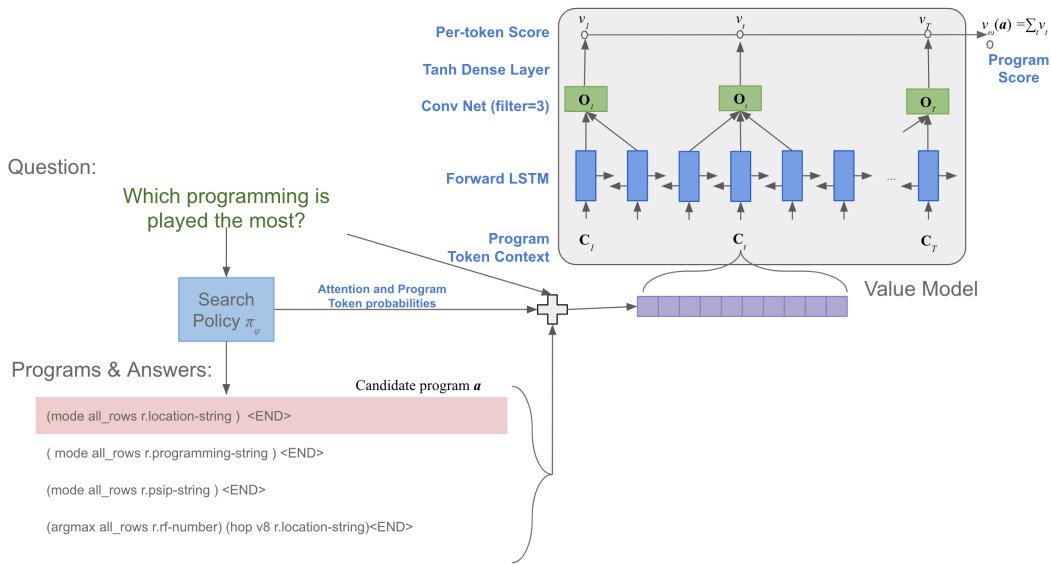

Figure 1: NPP architecture.

the program, question and beam context which are used to denote the utility of all tokens within a program (Section 4.2). Finally, we describe a training procedure that is based on max-margin/ranking objective on candidate programs given a question (Section 4.3).

## 4.1 ARCHITECTURE

Here we introduce the NPP architecture (Figure 1) in the context of semantic parsing, but the framework should be broadly applicable to applying RL in structured predictions. Given a pre-trained search policy $\pi_\phi$ and environment $\mathbf{x}$, NPP first unrolls the policy with beam search to generate candidate programs (plans) $\mathcal{B} = s_\phi(\mathbf{x})$. Then it scores each program $\mathbf{a}$ considering token $a_t$ at every step and global statistics among all programs $\mathcal{B}$. $a_t$ is represented as a context feature vector $\mathbf{C}_t$ (details in Section 4.2).

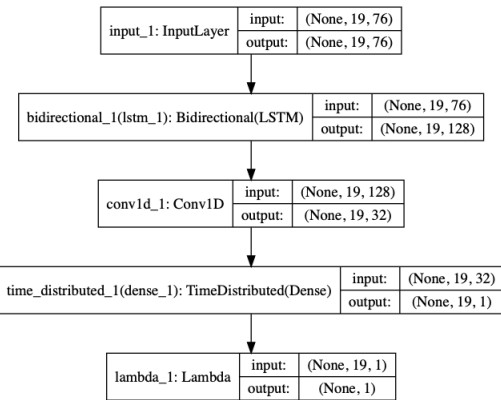

Figure 2: Value Model Details.

To capture the sequential inputs the scoring component is a seq2seq model which goes over candidate program answer pairs and assigns preference scores to each program/answer token. We implement a bi-directional recurrent network with LSTM Hochreiter & Schmidhuber (1997b); Greff et al. (2015) cells as the first layer of our planner $\mathbf{C}^{\text{LSTM}} = \text{LSTM}(\mathbf{C})$. The LSTM hidden state at each step is fed to a one dimensional convolutional layer with kernel size 3, in order to capture inner function scoring as most functions are of size 3-5, as a feature extractor $\mathbf{O}_t^{\text{CNN}} = \text{CNN}(\mathbf{C}_t^{\text{LSTM}})$. Finally we calculate the score per token by feeding into a single node hyperbolic tangent activation layer to compute the score per token $v_t = \text{Tanh}(\mathbf{O}_t^{\text{CNN}})$ of token $a_t$. The final score of a candidate

Table 2: Program Token Representation

| Symbol | Type | Meaning |
|---|---|---|
| $q^{\text{tok}}$ | binary | The program token matches any of the question tokens. |
| $\mathbf{q}^{\text{attn}}$ | float vector | Softmax attention over query tokens per program token |
| $p^{\text{prob}}$ | float | Program probability according to the search policy $\pi_\phi$ |
| $t^{\text{prob}}$ | float | Program token probability according to the search policy $\pi_\phi$ |
| $t^{\text{agree}}$ | count | Number of candidate programs having token $a_t$ at position $t$ |

program $v_\omega(\mathbf{a}) = \sum_{t=1..T} v_t$ is simply the sum of its token scores. The choice of simply summing token level scores makes the score very understandable (details in Section 4.2). Figure 2 gives implementation details of the value model.

## 4.2 PROGRAM TOKEN REPRESENTATION

To better score tokens based on the overall context of the environment we represent each token with a set of context features $\mathbf{C}_t = [q^{\text{tok}}; \mathbf{q}^{\text{attn}}; p^{\text{prob}}; t^{\text{prob}}; t^{\text{agree}}]$ as in Table 2. $\mathbf{q}^{\text{attn}}$ is the softmax attention across question tokens, which helps to discern which part of the question is of most importance to the model when generating the current given token. Together $q^{\text{tok}}$ and $\mathbf{q}^{\text{attn}}$ represents the alignment between program and query. $t^{\text{prob}}$ and $p^{\text{prob}}$ are the probability of token $a_t$ and program $\mathbf{a}$ assigned by the search policy $\pi_\phi$, which represent the decisions from the search model. $t^{\text{agree}}$ is the number of candidate programs having token $a_t$ at position $t$. Access to information such as $t^{\text{agree}}$ is only available to NPP as it can only be used after all the candidate programs have been generated.

## 4.3 TRAINING NPP

We formulate NPP training as a learning to rank problem — optimizing pairwise ranking among candidate programs $\mathcal{B}$. Given a training dataset of context-answer pairs, $\{(\mathbf{x}^l, \mathbf{y}^l)\}_{l=1}^N$, the training objective is

$$\mathcal{O}_{\text{NPP}}(\omega) = \sum_l \sum_{1 \leqslant i \neq j \leqslant |s_\phi(\mathbf{x}^l)|} \mathbb{1}[r^{l,i} > r^{l,j}] \log \sigma(v^{l,i} - v^{l,j}) \tag{2}$$

where $\sigma(v) = 1/(1 + e^{-v})$ is the sigmoid function, and $v^{l,i} = v_\omega(\mathbf{a}^{l,i}; \mathbf{x}^l, s_\phi(\mathbf{x}^l))$ is the estimated value of $\mathbf{a}^{l,i}$, the $i$-th program generated for context $\mathbf{x}^l$.

We compare NPP training in two setups: a single MAPO setup, and a stack learning setup. For the single MAPO setup the queries used to produce a pretrained MAPO model are also used to train the NPP model. The dev and test queries are used for ablation study and final evaluation. Since the NPP training queries are already used to train the MAPO model, the candidate programs are biased towards better reward (compared to those candidate generated for unseen queries). This setup causes NPP to learn from a different distribution as the intended dev/test data. Surprisingly NPP is still able to improve the prediction of MAPO as will be discussed in Section 5.3.

To overcome the issue with single MAPO setup we also generate NPP training data with a stacked learning setup. First the train and dev queries are merged and splitted into $K$ equal portions, and with Leave-One-Out (LOO) scheme they form $K$ train/dev sets. Then $K$ MAPO models are trained on $K$ train sets. Finally we use each of the $K$ MAPO s to produce a *stack learning dataset* by running these models on their respective dev set. The stack learning dataset is further splitted for train/dev purposes for NPP. In this way, each NPP training query is decoded by a MAPO model, which has never seen this query before, and therefore avoid the bias in training data generation.

## 5 EXPERIMENTS

Our empirical study is based on the semantic parsing benchmark, WIKITABLEQUESTIONS (Pasupat & Liang, 2015). To focus on studying the planning part of the problem we assume that the search policy is pretrained using MAPO (Liang et al., 2018), and NPP is trained to rescore given candidate programs produced by MAPO. Additionally we show that stacked learning (Cohen & Carvalho, 2005) is helpful in correctly training a planner given pre-trained policy models.

## 5.1 EXPERIMENTAL SETUP

**Datasets.** WIKITABLEQUESTIONS (Pasupat & Liang, 2015) contains tables extracted from Wikipedia and question-answer pairs about the tables. See Table 1 as an example. There are 2,108 tables and 18,496 question-answer pairs. We follow the construction in Pasupat & Liang (2015) for converting a table into a directed graph that can be queried, where rows and cells are converted to graph nodes while column names become labeled directed edges. For the questions, we use string match to identify phrases that appear in the table. We also identify numbers and dates using the CoreNLP annotation released with the dataset.

**Baselines.** We compare NPP to Liang et al. (2018), the current state of the art on the WIKITABLE-QUESTIONS dataset. MAPO relies on beam search to find candidate programs, and uses the program probability according to the policy to determine the program to execute. MAPO manages to achieve $46.3\%$ accuracy on this task when using an ensemble of size 10. We aim to show that NPP can improve on single MAPO as well as the ensemble of MAPO models.

**Training Details.** We set the stacked learning parameter $K = 5$ for all our experiments. We set the batch size to be equal to 16. We use Adam optimizer for training with a learning rate $10^{-3}$. We choose a size of 64 nodes for the LSTM (which becomes 128 as it is bidirectional). The CNN consists of 32 filters with kernel size 3. All the hyper parameters are tuned on the development set and trained for 10 epochs.

**Ensemble Details.** We formulate the ensemble of $K$ MAPO models with NPP as the sum of normalized NPP scores under different MAPO models: Let $\Phi = \{\phi^k\}_{k=1}^K$ be $K$ MAPO models to be ensembled. We define the ensembled score of a program $\mathbf{a}$ under context $\mathbf{x}$ as

$$v_{\omega,\Phi}(\mathbf{a};\mathbf{x}) = \sum_k [v'_\omega(\mathbf{a};\mathbf{x}, s^k_\phi(\mathbf{x})) - \bar{v}_\omega(\mathbf{x})]. \tag{3}$$

where $\bar{v}_\omega(\mathbf{x})$ is the average score for programs in beam $s^k_\phi(\mathbf{x})$

$$\bar{v}_\omega(\mathbf{x}) = \frac{1}{|s^k_\phi(\mathbf{x})|} \sum_{\mathbf{a} \in s^k_\phi(\mathbf{x})} v_\omega(\mathbf{a};\mathbf{x}, s^k_\phi(\mathbf{x})) \tag{4}$$

and $v'_\omega$ backs-off $v_\omega$ to $\bar{v}_\omega(\mathbf{x})$ whenever $\mathbf{a}$ is not in beam

$$v'_\omega(\mathbf{a};\mathbf{x}, s^k_\phi(\mathbf{x})) = \begin{cases} v_\omega(\mathbf{a};\mathbf{x}, s^k_\phi(\mathbf{x})), & \text{if } \mathbf{a} \in s^k_\phi(\mathbf{x}) \\ \bar{v}_\omega(\mathbf{x}), & \text{else} \end{cases} \tag{5}$$

Table 3: Feature ablation study on the dev set with a mean of 5 runs on a single MAPO setup.

| Setting | $\Delta$ Acc(std) |
|---|---|
| NPP- $q^{\text{tok}}$ | $-0.37(0.04)$ |
| NPP- $\mathbf{q}^{\text{attn}}$ | $-0.04(0.2)$ |
| NPP- $p^{\text{prob}}$ | $-6.04(0.7)$ |
| NPP- $t^{\text{prob}}$ | $-0.14(0.03)$ |
| NPP- $t^{\text{agree}}$ | $-0.39(0.2)$ |

## 5.2 ABLATION STUDY

In order to evaluate the effectiveness of our proposed programs token representations, we present a feature ablation test in Figure 3. We can see that the program probability $p^{\text{prob}}$ produced by the search policy is the most important feature, providing the foundation to NPP scoring.

The program agreement feature $t^{\text{agree}}$ is also very useful. We often observe cases for which beam $\mathcal{B}$ produces program with similar tokens that are not highly valued by the underlying model. By utilizing this feature, we more strongly consider programs which were repeatedly searched by $s_\phi$. $t^{\text{agree}}$ also helps to identify the programs that are very similar throughout most of the sequence to learn their divergence and grade their utility.

Table 4: Main results. [†]Improvements compared to MAPO. [*]Stacked learning with Leave-One-Out (LOO) data splits. [+]NPP uses 67%-33% train-dev splits from the stacked learning data.

| Setting | Model | Dev (std) | $\Delta^\dagger$ | Test (std) | $\Delta^\dagger$ |
|---------|-------|-----------|------------------|------------|------------------|
| Mean of MAPOs trained on a single train/dev split | MAPO | 41.9(0.3) | - | 43.1(0.5) | - |
| | MAPO + NPP | 42.4(0.7) | 0.8 | 43.7(0.6) | 0.5 |
| Mean of MAPOs trained on LOO splits | MAPO | 41.7(1.1) | - | 42.8(0.5) | - |
| | MAPO + NPP* | $43.0(0.2)^+$ | 1.3 | 43.9(0.2) | 1.1 |
| Ensemble of 5 MAPOs trained on LOO splits | MAPO | - | - | 45.5 | - |
| | MAPO + NPP* | - | - | 46.6 | 1.1 |
| Ensemble of 10 MAPOs trained on LOO splits | MAPO | - | - | 46.3(−) | - |
| | MAPO + NPP* | - | - | 47.2(−) | 0.9 |

Question referencing features such as $q^{\text{tok}}$ provide significant importance in providing the program with query level context, ensuring we are filtering or selecting values based on the query context. While the help from attention feature $\mathbf{q}^{\text{attn}}$ is not significant.

## 5.3 Main Results

We evaluate NPP's impact on MAPO under three different settings, in each of which NPP consistently improves the precision of MAPO.

First we consider a single MAPO trained on a single train-dev data split. Similar to other RL models in the literature, MAPO training is a procedure with big variances. Even trained on the same data split multiple times with different random seed gives big variances in accuracy of 0.3% (dev) and 0.5% (test). We use the MAPO model to decode on its own train, dev and test data, in order to generate parallel splits for NPP training. Training and applying NPP this way is able to improve precision, despite of the exposure bias in the NPP training data. However, it does not improve on the variances.

We next consider MAPO models that were trained and evaluated on separate train/dev splits created with a Leave-One-Out (LOO) scheme. As described in Section 4.3 we also use these splits to generate a stacked learning dataset for NPP to avoid the data bias problem. We can see that with different data splits MAPO has significantly higher variances on the dev set (1.1%), which is a drawback of RL models in general. Stacked learning helps NPP to improve precision more significantly (1.3% for dev and 1.1% for test). It also helps to reduce the variances to 0.2% on both dev and test.

Finally, we consider ensembled MAPO settings, which produces the previous state of the art result. We use the same NPP model trained from the stacked learning setting, and apply it to an ensemble of either 5, or 10 MAPO models from (Liang et al., 2018). We can see that when applied to 5 MAPO ensemble, NPP can still improve the precision by 1.1%. When applied to 10 MAPO ensemble, NPP can improves the precision by 0.9%.

## 5.4 Explainability Analysis

Since the score of a program is the sum of its token scores, it is easy to visualize how NPP plan and select the correct program to execute. We observed that there are two common situations in which MAPO chooses the wrong program from the beam – selecting a spurious program over the semantically correct program and executing the incorrect table filtering or column selection. NPP aims to overcome these non optimal decisions by taking advantage of the larger time horizon and other programs discovered so far. For example NPP may reward earlier program tokens based on program tokens chosen much later on due to the bi-directional recurrent network.

We first investigate how NPP demotes spurious programs. MAPO may produce programs which return the correct answer but are not semantically correct. NPP helps solve this by scoring semantically correct programs higher in the beam. An example is shown in Figure 3 for the question

*"What venue was the latest match played at?"* when referring to a soccer (football) player given a table of his matches. The top program in beam proposed by MAPO was to first filter all rows for the competition string, which is incorrect considering the context of the table is only competitions. NPP is able to reevaluate the program given the full context. Although the first function (**filter_in**) is typically used to filter the table for the correct row/column. NPP learns that in this situation it is better to find the last of all rows using the function **last**. NPP, re-evaluates the first function of the new best program as being high in utility, and scores all tokens within this function higher than the tokens in the incorrect program.

We then investigate programs from MAPO which produce wrong answers. An example is shown in Figure 4 which is based on Table 1. MAPO assigns a higher probability to a program in beam that filters on an incorrect column. Because NPP knows program token overlap with the query as well as the attention matrix, it is able to better understand the question and grade the program which is more closely related to the question. In addition to this we notice that the convolutional layer grades full functions within their context, given a kernel of size 3 the parenthesis at the beginning of program already receives a higher NPP score which we interpret as the overall score of executing the function.

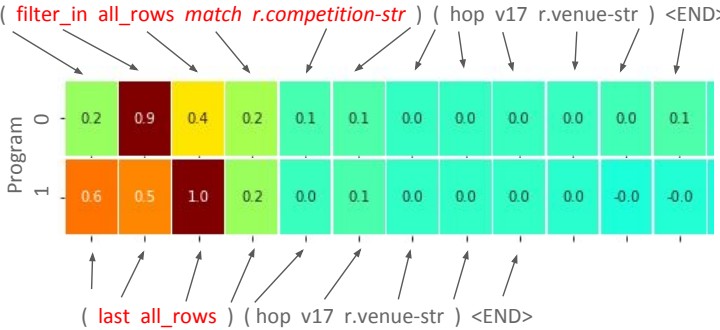

Figure 3: Comparison of re-scorings by NPP. (Top) a semantically incorrect program now scored lower by NPP. (Bottom) a semantically correct program now scored higher by NPP. **nt-10858**: *What venue was the latest match played at?*

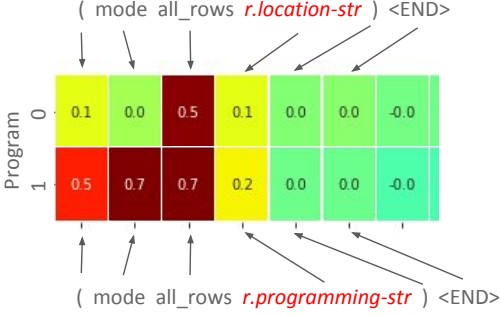

Figure 4: Comparison of re-scorings by NPP. (Top) a value incorrect program now scored lower by NPP. (Bottom) a value correct program scored higher by NPP. **nt-3516**: *Which programming is played the most?*

## 6 CONCLUSION

Reinforcement learning applied to structured prediction suffers from limited use of the world model as well as not being able to consider future and past program context when generating a sequence. To overcome these limitations we proposed Neural Program Planner (NPP) which is a planning step after candidate program generation. We show that an additional planning model can better evaluate overall structure value. When applied to a difficult SP task NPP improves state of the art by 0.9% and allows intuitive analysis of its scoring model per program token.

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

## A    MORE NPP SCORING DETAILS

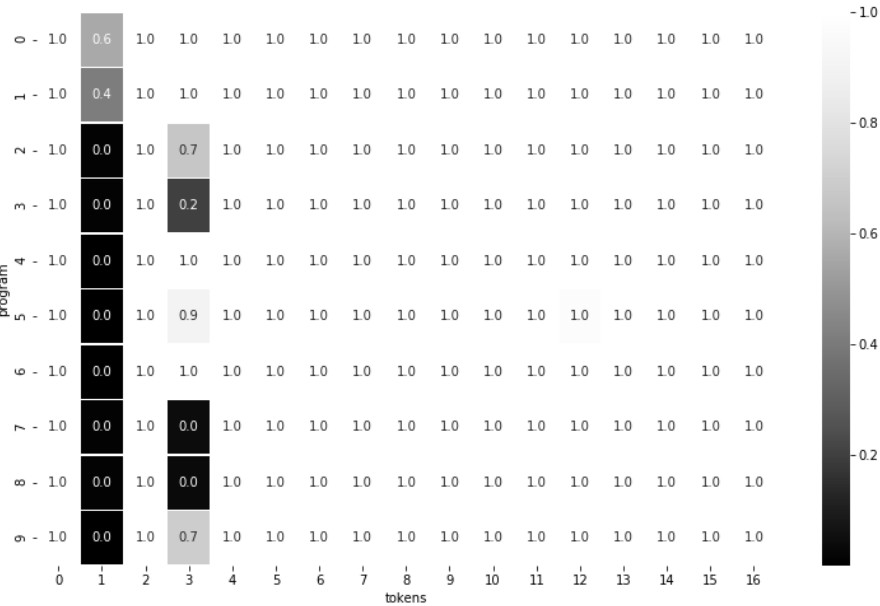

Figure 5: MAPO probability per token for **nt-10858**: *What venue was the latest match played at?* The score sequences have the same length (16) because of padding within this query.

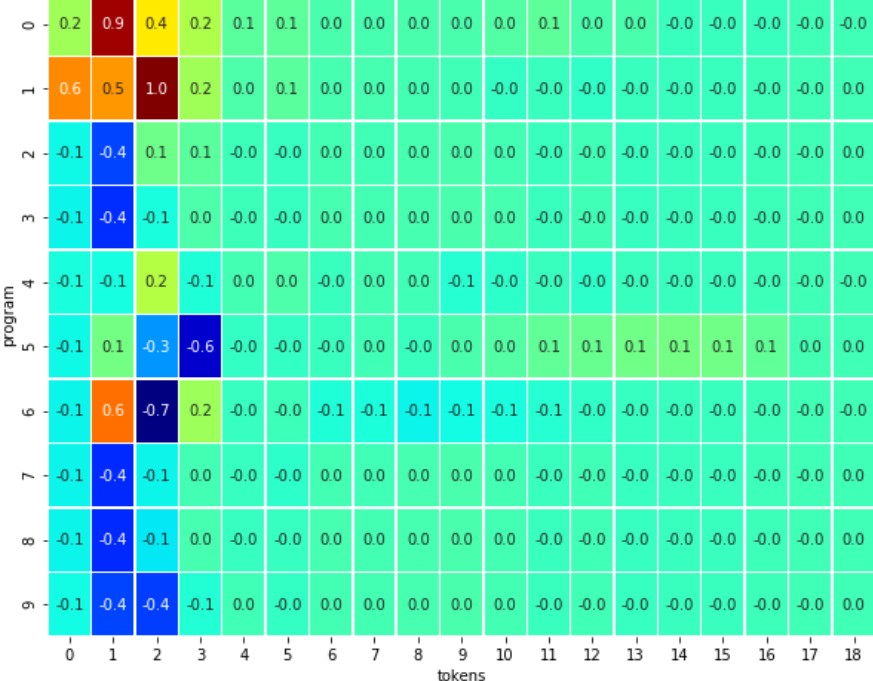

Figure 6: NPP scores per token for a set of candidate programs. **nt-10858**: *What venue was the latest match played at?* The score sequences have the same length (19) because of padding over all queries.

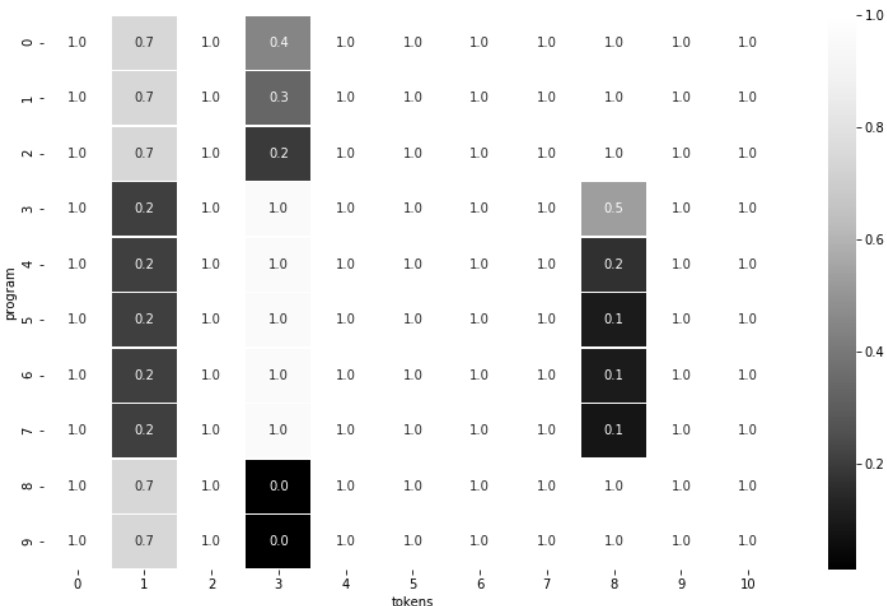

Figure 7: MAPO probability per token for **nt-3516**: *Which programming is played the most?* The score sequences have the same length (10) because of padding within this query.

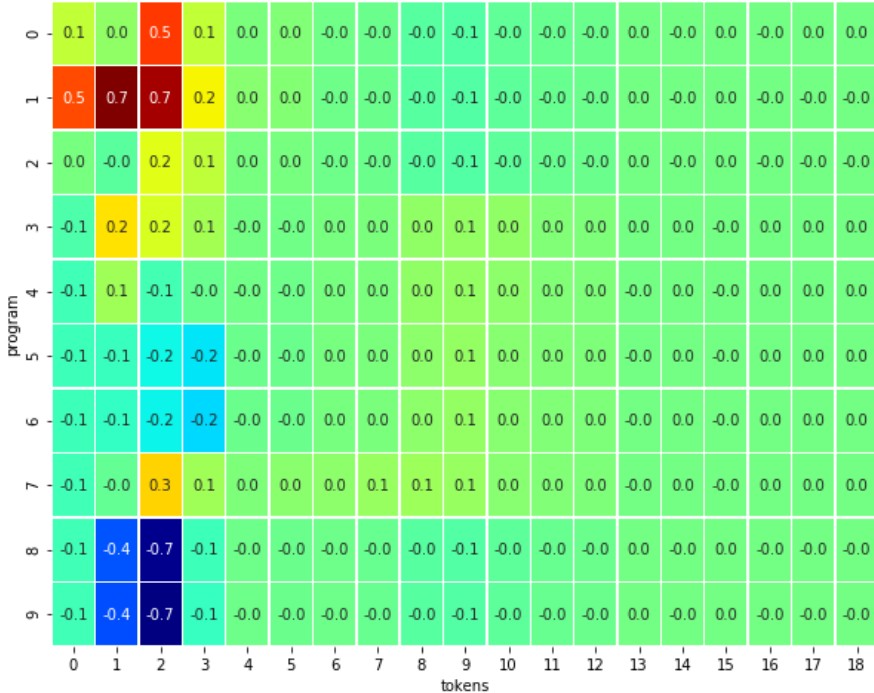

Figure 8: NPP scores per token for a set of candidate programs. **nt-3516**: *Which programming is played the most?* The score sequences have the same length (19) because of padding over all queries.

