# OpenReview forum: "Neural Program Planner for Structured Predictions"
_ICLR.cc/2019/Workshop/drlStructPred — drlStructPred 2019_

### Official Review · AnonReviewer3 · 2019-04-04
**Good motivation and experimental setup but results are not convincing enough.**

**Rating:** 2
**Confidence:** 2

**Review:**

The paper proposes a re-ranking function to improve program generation on the task of answering questions that use tables as context.

Pros:
1- Interesting task that deserves our attention because of its practical applications.
2- The ablation study nicely justifies the extra inputs.

Cons:
1- The paper is hard to understand. Also, it contains some typos and grammatical errors.
2- 0.9-1.1% is a small improvement over MAPO.

Questions:
1- If the final score given by NPP is the sum of token values in the candidate program, how do you prevent longer programs to have higher scores?
2- Did you have to subtract the reward by a baseline to train the search policy with REINFORCE?

---

### Official Review · AnonReviewer4 · 2019-04-05
**RL-inspired re-ranker in beam search leads to better structure prediction results in one task: WikiTable QA. Writing needs substantial improvement.**

**Rating:** 2
**Confidence:** 2

**Review:**

The authors propose to improve the quality of beam search in structure prediction by using RL-inspired techniques for learning a forward-cost estimator; the forward-cost estimator is used to re-rank hypotheses in a typical beam search. The proposed method is shown to improve the state of the art on the WikiTable Question Answering task [Liang et al, 2018].

Although the results are interesting, it is not clear to what extent the improvements are explained by the use of RL-inspired techniques or the fact that the forward-cost estimator accesses additional sources of knowledge. These additional sources of knowledge need to be available at training/decoding time, so the approach amounts to a simple re-ranking of hypotheses surfaced via typical beam search – as a consequence, the approach may help re-rank search results that are already encoded in the beam, but not search results that fell off the beam.

The paper has some interesting ideas, but the writing is very poor. A substantial rewriting of the submission that goes beyond documenting the authors’ train of thought would help readers understand better the main claims and limitations of the approach.

---

### Official Review · AnonReviewer2 · 2019-04-06
**Interesting idea, good results, clarity and presentation needs improvement**

**Rating:** 3
**Confidence:** 2

**Review:**

Pro:
+ Neural Program Planner is proposed to evaluate the value of generations by considering structural and context information.
+ State-of-the-art is achieved on a benchmark dataset WikiTableQuestions.

Con:
- The presentation can be improved, in terms of writing and model explanation.
- It's unclear if the summation of token-level scores is the best way for aggregation. It's possible to learn another function over the outputs from conv nets (Os in figure 1).

Other comments:
The first time NSM and MAPO are introduced, they should be spelled out.
Typos, "considering the its program", "sequence to sequence model" (should be sequence-to-sequence model)...

---

### Decision · Program_Chairs · 2019-04-09
**Acceptance Decision**

**Decision:**

Accept

**Comment:**

Even though the results are very preliminary we still accept them for the purpose of fostering interesting discussions.